
# Dynamic Risk Assessment of Compound Hazards Based on VFS-IEM-IDM: A Case Study of Typhoon-Rainstorm Hazards in Shenzhen, China

Wenwu Gong[1], Jie Jiang[2], and Lili Yang[3]

[1,3]Department of Statistics and Data Science, Southern University of Science and Technology, Shenzhen, 518055, China
[2]Department of Computer Science and Engineering, Southern University of Science and Technology, Shenzhen, 518055, China

**Correspondence:** Lili Yang (yangll@sustech.edu.cn)



**Abstract.** Typhoons and rainstorms are types of natural hazards that can cause significant impacts. These individual hazards may also occur simultaneously to produce compound hazards, leading to increased losses. The accurate risk assessment of such compound hazards faces several challenges due to the uncertainties in multiple hazards level evaluation, and the incomplete information in historical data sets. In this paper, to deal with these challenges, we propose a risk assessment model called

VFS-IEM-IDM based on the Variable Fuzzy Set and Information Diffusion Method. In particular, VFS-IEM-IDM provides a comprehensive evaluation of the compound hazards level, and a predictive cumulative logistic model is used to verify the results. Furthermore, VFS-IEM-IDM applies a normal information diffusion estimator to estimate the conditional probability distribution and the vulnerability distribution of the compound hazards based on the hazards level, the hazards occurrence time, and the corresponding losses. To examine the efficacy of VFS-IEM-IDM, a case study of the Typhoon-Rainstorm hazards that

occurred in Shenzhen, China is presented. The risk assessment results indicate that hazards of level II mostly occur in August and October, while hazards of level III often occur in September. The risk of the Typhoon-Rainstorm hazards differs in each month and in August and September the risk gets the highest value, and the estimated economic losses are around 114 million RMB and 167 million RMB respectively.

**Key words:** Compound hazards risk; Fuzzy dynamic risk; Variable fuzzy set; Information diffusion; Typhoon-Rainstorm



## 1  Introduction

Assessing risk is an effective way to reduce the negative impacts on natural hazards and plays an increasingly important role in helping the decision maker in emergency management. With the global climate change, many cities have suffered extreme natural hazards more frequently and many people's lives are under threat. Located in the southern part of China, Shenzhen is a coastal city with a low latitude, where Typhoon and Rainstorm hazards have severely restricted the sustainable development of
the local economy and society. Furthermore, the development of the Guangdong-HongKong-Macao Greater Bay Area highly relies on timely and effective emergency plans which are often determined by the efficiency of the risk assessment.

Risk assessment is a technique that uses hazards data to estimate the probability that natural hazards occur and assess their economic losses. Traditional methods of risk assessment mainly utilize Geographic Information System (GIS) to get risk maps (Gigovic et al. (2017)), or rely on information diffusion method (IDM) to deal with incomplete data sets (Gong et al. (2020)).
These relevant risk assessment methods (Julia et al. (2021); Zhou et al. (2020)) have became more comprehensive and mature in single hazards evaluation. However, the multi-hazard risk assessment is not the aggregation of their individual assessment results but considers the connections among different hazards (Kappes et al. (2012)), so the assessment results for multiple hazards are often inaccurate and insufficient. Furthermore, there is little research focusing on Typhoon-induced risk assessment in the literature and many aspects such as dynamic risk assessment are not considered.


There are many works discussing the multi-hazard risk assessment and Choi et al. (2021) had reviewed the relevant literature. Furthermore, Wang et al. (2020) clarified the relationship between hazards in multi-hazard scenarios by dividing them into three categories: mutually amplified hazards, mutually exclusive hazards, and non-influential hazards. Khan et al. (2020) presented an analysis of the existing methods and technologies that are relevant to multi-hazard scenarios. Huang et al. (2018) developed information diffusion technique to construct a joint probability distribution and a vulnerability distribution
for assessing the flood and earthquake risks. Xu et al. (2016) also used the information diffusion method to assess the risk of multiple hazards quantitatively and evaluated the risk of loss of human lives from meteorological hazards in China. Ming et al. (2015) proposed a quantitative approach of multi-hazard risk assessment based on vulnerability distribution and joint return period of hazards to assess the risk of crop losses in the Yangtze River Delta region of China. However, all of these works focus on integrating the risks caused by single hazards and ignoring these truly correlations between hazards occurrence, such
as the co-appearance of Typhoon-Rainstorm hazards. In this paper, we aim at the multi-hazards and investigate new methods for multiple hazards level evaluation and dynamic risk assessment of compound hazards.

Compound hazards, a sub-group of the term 'multi-hazard', can be considered as the combination of multiple drivers that contribute to societal risk (Jennifer et al. (2021)), within which two associated hazards impacting the same time and place. In this paper, we define the compound hazards risk as a scene in the future associated with some adverse incident caused
by cascading hazards systems, where there are strong connections among different hazard indicators. Compared with the multi-hazard risk assessment (Xu et al. (2016); Huang et al. (2018)), the risk assessment of compound hazards obtains the comprehensive hazards level without losing any correlated information and often reflects the property of hazard-induced. Risk assessment of compound hazards has been studied by He et al. (2020), who presented the Cloquet integral multiple





linear regression model to overcome the problems of nonlinear additivity of couple hazards. But this method only provides
the magnification coefficients to assess the risks of compound hazards and neglects the changing of time span. Here, there
are some problems remaining to be solved. On the one hand, the collected data for assessing the compound hazards risk is
often incomplete such that the results may not be reliable. On the other hand, the change of month in which the compound
hazards occur also has impacts on the risk assessment and is often ignored. In this paper, we emphasize that risk assessment
of compound hazards should deal with the uncertainties caused by multi-indicators, the unknown probability distributions, the
incomplete information in historical data sets, and the dynamic property of hazards occurrence.

Some research based on variable fuzzy sets (VFS) theory, introduced by Chen (2006), have shown that the relative membership function can be used to evaluate the multi-indicators assessment problems. Li et al. (2012) proposed the fuzzy comprehensive assessment method to solve the flood risk assessment problems with interval boundaries and this integrated model
improves the reliability of single hazard risk assessment. Beaula et al. (2013) used variable fuzzy sets to evaluate the synthetic
hazards level of Nagapattinam district with the north-east monsoon rainfall's data sets. Similarly, the variable fuzzy set theory
can be used to obtain the comprehensive evaluation of compound hazards. In this paper, we propose to combine the VFS with
information entropy method (IEM) to assess compound hazards indicators and obtain a comprehensive risk assessment.

In many cases, it is difficult to collect compound hazards data sets, such that the information carried by historical data is often
incomplete. Therefore, the traditional models often give an unreliable estimation result, and many fuzzy probabilistic models
have been proposed to enhance the accuracy of risk assessment Mehran et al. (2017). Fuzzy probabilistic models are used
to model uncertainties related to hazards and the randomness due to environmental, natural, or time span changing. The main
feature of the fuzzy probabilistic model is to change the traditional data points into fuzzy set for partly filling the gap caused
by data incompleteness and improve the estimation accuracy between inputs and outputs. The most powerful technique is the
information diffusion method (IDM), which helps extract useful underlying information from the hazard data sets. Research by
Huang (1997, 2002) has given many results about IDM and there are many papers have shown the capability of information
diffusion method to deal with incomplete data sets (Huang (2009); Li (2013); Huang et al. (2018). In this paper, we introduce
the information diffusion method to deal with the incomplete data problem and combine the variable fuzzy sets theory to carry
out dynamic risk assessments of compound hazards.

The main contributions of this paper are summarized as follows.

– 1) We consider the uncertainties in compound hazards level evaluation and incomplete information in historical data
       sets, and propose a hybrid model, named as Variable Fuzzy Set and Information Diffusion Method (VFS-IEM-IDM), to
       deal with compound hazards risk assessment dynamically.

    – 2) To improve the efficiency and accuracy of compound hazards level evaluation, the calculation procedures of relative
       membership degree have been categorized into three types.

– 3) To examine the efficacy of VFS-IEM-IDM, a case study of the Typhoon-Rainstorm hazards that occurred in Shenzhen,
       China is presented.





The rest of this paper is organized as follows. Section 2 introduces the basic concepts and presents the dynamic compound hazards risk assessment model (VFS-IEM-IDM). Section 3 illustrates how the proposed model can be used to assess the dynamic risk of Typhoon-Rainstorm hazards in Shenzhen, China. Section 4 discusses the comprehensive evaluation of the

compound hazards level, conditional probability distribution, vulnerability distribution and the dynamic expectation risk of the Typhoon-Rainstorm hazards to show the effectiveness of VFS-IEM-IDM. Finally, we conclude the paper in Section 5.

## 2   Dynamic Risk Assessment of Compound Hazards Based on VFS-IEM-IDM

Risk assessment of compound hazards should consider the uncertainties caused by multi-indicator, incomplete information contained in historical data sets, and the impact of internal attribute changes on the hazards. This section introduces VFS-

IEM-IDM which combines the variable fuzzy sets theory with information diffusion method to assess the dynamic risk of compound hazards when the given data sets are incomplete. The proposed VFS-IEM-IDM model consists of VFS-IEW dimension reduction model to obtain the comprehensive evaluation of compound hazards level (Section 2.2), and VFS-IDM dynamic risk assessment model to estimate the expectation risk of direct economic losses (Section 2.3).

### 2.1   Dynamic compound hazards risk

Risk is assumed to be the possible scene of the occurrence of a harmful event. From the previous studies, the type of risk could be classified into four categories: pseudo risk, probability risk, fuzzy risk, and uncertainty risk (Huang et al. (2018)). In the case of that we can estimate the probability distribution $p(x)$ (hazard potential) of the occurrence of a hazard with respect to its magnitude $x$, and we can estimate the relationship $f(x)$ (hazard vulnerability) between the magnitude and hazard level, a probability risk could be quantified as the expected value of economic losses, i.e.,

$$Risk = Hazard\ Potential \times Vulnerability. \tag{1}$$

Though these four types of risks have been investigated by many researchers, there is little research on dynamic compound hazards risk. In this paper, we give a definition of dynamic compound hazards risk and illustrate how to assess this kind of risks.

*The compound risk* is a scene in the future associated with some adverse incident caused by cascading hazards systems,

where there are strong connections between different hazards and the hazard level is influenced by many indicators. Furthermore, Huang (2015) mentioned that it could extent to *dynamic compound risk* if the impact of occurrence time on risk assessment has been taken into consideration. To evaluate the compound hazards risk, the most important things are to estimate probability distribution $p(x)$ of the occurrences of compound hazards by using probability models, and the input-output relationship $f(x)$ between the hazards level and losses by using fuzzy models. The compound risk, quantified as the economic

losses of compound hazards, is given by Eq. 2.

$$Risk = \int p(x) \cdot f(x)\, dx = \sum_{j=1}^{J} p(x; att_j) \cdot f(x; att_j), \tag{2}$$





where vector $att_j$ denotes the hazards indicator for different index $j$ and reflects the internal attribute changes of compound hazards. For example, the Typhoon-Rainstorm risk is influenced by different indicators $att$ = (compound hazards level, economic losses) and the dynamic compound risk can be assessed by integrating conditional probability distribution $p(x; att)$ with hazards vulnerability $f(x; att)$ of Typhoon-Rainstorm hazards.

## 2.2 VFS-IEW dimension reduction model

For the compound hazards risk assessment system, the randomness and fuzziness caused by multi-indicators evaluation should be dealt with properly. Variable fuzzy sets theory (VFST), which deals with randomness and fuzziness, provide an appropriate tool for solving the compound hazards level evaluation. The Variable Fuzzy Sets-Information Entropy Weight (VFS-IEW) dimension reduction model has been proposed in this section.

In this paper, we define interval $I_0 = [a, b]$ as the attracting sets of variable fuzzy sets (VFS) $U$ and extends $I_0$ to interval $I = [c, d]$ on the real axis. For $u \in U$, the elements in interval $I_0$ satisfy $\mu_A(u) > \mu_A^c(u)$. In VFST, $\mu_A(u)$ denotes the relative membership degree (RMD) and the core idea is to determine the RMD of each sample point by transferring fuzzy set $U$ into real value. Wang et al. (2014) has defined the balance boundaries matrix and illustrates the calculation of RMD as a complicated and time consuming problem. We apply the balance boundaries matrix $M = \{M_{rl}\}$ (shown in Eq. 3) to locate the eigenvalue $x$ and defines the relative membership degree functions (shown in Eq. 4) to evaluate the comprehensive value of compound hazards level.

$$M = \frac{L-l}{L-1}a_{rl} + \frac{l-1}{L-1}b_{rl} = (M_{rl}), \tag{3}$$

where $r$ stands for the assessment indicator set, $r = 1, 2, \ldots, R$, $l$ denotes the comprehensive level, $l = 1, 2, \ldots, L$. Compared with relative locations of sample points and parameter $M_{rl}$, RMD calculation can be solved by the ratio $\frac{x-a}{M-a}$, i.e.

$$\begin{cases} \mu_A(u) = 0.5[1 + \left(\frac{x-a}{M-a}\right)^p] & x \in [a, M] \\ \mu_A(u) = 0.5[1 - \left(\frac{x-a}{c-a}\right)^p] & x \in [c, a] \end{cases}. \tag{4}$$

It can be seen that the RMD is affected by hyper-parameter $p$ and the position between sample point $x$ with parameters a, b, c, d, and $M$. In this paper, the characteristics for different locations of $x$ with respect to the class interval $u$ have been used to classify RMD calculation: judge whether the location of $x$ is in the lowest or highest grade of the class interval $u$ or not. Fig.1-Fig.3 have shown three types of RMD calculation and the detailed induction can be referred by Fang et al. (2019).

$$\begin{cases} \mu_A(u)_1 = [\mu_A(u)_{11} \quad \mu_A(u)_{12} \quad 0 \quad \cdots \quad 0] \\ \mu_A(u)_{11} + \mu_A(u)_{12} = 1 \\ 0.5 \leq \mu_A(u)_{11} \leq 1 \\ 0 \leq \mu_A(u)_{12} \leq 0.5 \end{cases}. \tag{5}$$


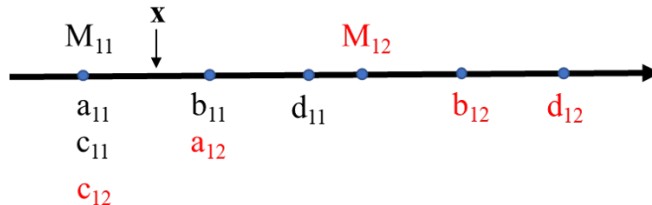

**Figure 1.** Lowest: the position between the random point $u_1{}^t = x$ with parameter $M_{11}$ and zones $[a_{11}, b_{11}]$, $[c_{11}, d_{11}]$.

**Figure 2.** Highest: the position between the random point $u_1{}^t = x$ with parameter $M_{1L}$ and zones $[a_{1L}, b_{1L}]$, $[c_{1L}, d_{1L}]$.

$$\begin{cases} \mu_A(u)_1 = \begin{bmatrix} 0 & \cdots & 0 & \mu_A(u)_{1(L-1)} & \mu_A(u)_{1L} \end{bmatrix} \\ \mu_A(u)_{1(L-1)} + \mu_A(u)_{1L} = 1 \\ 0.5 \le \mu_A(u)_{1L} \le 1 \\ 0 \le \mu_A(u)_{1(L-1)} \le 0.5 \end{cases} \tag{6}$$

**Figure 3.** Middle: the position between the random point $u_1{}^t = x$ with parameter $M_{1l}$ and zones $[a_{1l}, b_{1l}]$, $[c_{1l}, d_{1l}]$.





$$\begin{cases} \mu_A(u)_1 = [0 \quad \cdots \quad 0 \quad \mu_A(u)_{1(l-1)} \quad \mu_A(u)_{1l} \quad \mu_A(u)_{1(l+1)} \quad 0 \quad \cdots \quad 0] \\ \mu_A(u)_{1(l-1)} + \mu_A(u)_{1(l+1)} = 0.5 \\ 0 \le \mu_A(u)_{1(l-1)} \le 0.5 \\ 0 \le \mu_A(u)_{1(l+1)} \le 0.5 \end{cases} . \tag{7}$$

Following the previous works by Kwakernaak (1978) and Chen (2006), the proposed variable fuzzy set dimension reduction

model can be constructed by Eq. 8. It indicates that the proposed model is affected by hyper-parameter $\alpha, \beta$ and the multi-

indicators are transferred into a single degree value so as to obtain the comprehensive assessment results.

$$\begin{cases} \nu_A(u)_l = [1 + (\frac{\sum_{r=1}^{R}[\omega_r(1-\mu_A(u)_{rl})]^\alpha}{\sum_{r=1}^{R}[\omega_r \mu_A(u)_{rl}]^\alpha})^{\frac{\beta}{\alpha}}]^{-1} \\ \nu_A^o(u)_l = \frac{\nu_A(u)_l}{\sum_{l=1}^{L}\nu_A(u)_l} \\ H = (1 \quad 2 \ldots L) \cdot (\nu_A^o(u)_l)^T \end{cases} . \tag{8}$$

where $\nu_A^o(u)$ is the normalized process of RMD and $H$ is the comprehensive value (a real value, can be transferred to hazards

level). Further, the weight of indicators in this VFS-IEW model can be calculated by information entropy weight (Liu et al.

(2010)) (shown in Eq. 9).

$$\begin{cases} f_r{}^t = u_r{}^t / \sum_{t=1}^{T} u_r{}^t \\ h_r = -1/lnn \cdot \sum_{t=1}^{T}(f_r{}^t ln f_r{}^t) \\ \omega_r = (1-h_r)/(R - \sum_{r=1}^{R} h_r) \end{cases} . \tag{9}$$

We now present the main steps of VFS-IEW model and the corresponding algorithm (shown in Algorithm 1) as follows.

- **Step-1:** Initialize the variable fuzzy sets and the balance boundaries $M_{rl}$.

- **Step-2:** Repeat the relative membership degree calculation.

– **Step-3:** Calculate the information entropy weight $\omega_r$.

- **Step-4:** Return the comprehensive degree value.

## 2.3   Dynamic risk assessment model

In order to assess the dynamic risk of compound hazards, especially when the recorded data sets are incomplete, information

diffusion method (IDM) which belongs to fuzzy sets theory can be used to extract useful underlying information from the sam-

ples to estimate the relationships behind the incomplete data. According to the research by Huang (1997), normal information




---

**Algorithm 1** VFS-IEW Dimension Reduction Model for Compound Hazards

---

**Input:**

1: The assessment object set $D = \{U_t = (u_r)^t, r = 1, 2, \cdots, R | t = 1, 2, \cdots, T\}$, where $u_r$ is the eigenvalue;

2: Assessment criteria matrix, $V = \{(v_{rl}), r = 1, 2, \cdots, R; l = 1, 2, \cdots, L\}$.

**Output:**

Comprehensive value of compound hazards level.

3: Identification of attracting sets $I_{ab} = ([a, b]_{rl})$ and the extended intervals $I_{cd} = ([c, d]_{rl})$ based on assessment criteria matrix $V$;

4: Define the balance boundaries matrix $M = \{(M_{rl}), r = 1, 2, \cdots, R; l = 1, 2, \cdots, L\}$ by Eq. 3;

5: Calculate the information entropy weight $\omega_r$ by Eq.9;

6: **for** $t = 1$ to $T$, each $U_t$ **do**

7:     **for** each $u_r{}^t \in U_t$ **do**

8:         **if** $u_r{}^t$ locates in the lowest grade of the class interval $I_{ab}$, i.e., $a_{r1} < u_r{}^t < b_{r1}$ **then**

9:             RMD $\mu_A(u)_r{}^t$ has the expression given by Eq.5;

10:         **else if** $u_r{}^t$ locates in the highst grade, i.e., $a_{rL} < u_r{}^t < b_{rL}$ **then**

11:             The value of RMD $\mu_A(u)_r{}^t$ is given by Eq.6;

12:         **else**

13:             RMD $\mu_A(u)_r{}^t$ have the expression given by Eq.7;

14:         **end if**

15:     **end for**

16:     The relative membership matrix of each sample can be denoted as $\mu_A(u)^t = (\mu_A(u)_{rl}{}^t)$;

17:     Combine $\mu_A(u)^t$ with weights $\omega_r$ and integrate the ranking level, the comprehensive degree value for each sample is given by Eq.8.

18: **end for**

---

diffusion function $\mu(X^t{}_i; S_o)$ (shown in Eq. 10) is more powerful to improve the precision of estimators. So, this paper adapts normal information diffusion estimator to approximate the dynamic compound hazards risk as follows.

$$\mu(X^t{}_i; S_o) = \prod_{o=1}^{3} exp[-\frac{(x_{ot} - s_o)^2}{2h_s^2}], \; i = 1, \cdots, n; \; t = 1, 2, \cdots, T. \tag{10}$$

$$h = \begin{cases} 0.6841(b - a), & for \quad n = 5; \\ 0.5404(b - a), & for \quad n = 6; \\ 0.4482(b - a), & for \quad n = 7; \\ 0.3839(b - a), & for \quad n = 8; \\ 2.6581(b - a)/(n - 1), & for \quad n \geq 9. \end{cases} \tag{11}$$

$$where \quad b = \max_{1 \leq i \leq n}\{x_i\}, \quad a = \min_{1 \leq i \leq n}\{x_i\}.$$




where $T$ is the different month value, $S_o$ denotes monitor space, and $h_s$ is the diffusion coefficient. Based on this normal estimator, the research by Huang (2002) has shown how to determine the coefficients (shown in Eq. 11) and the discrete probability density function can be estimated by matrix $P = \{p_{jk}\}$.

$$p_{jk} = \frac{\sum_{i=1}^{n} \mu(X^t{}_i; u_j, v_k)}{\sum_{j=1}^{J} \sum_{k=1}^{K} \sum_{i=1}^{n} \mu(X^t{}_i; u_j, v_k)}, j = 1, 2, \cdots, J; k = 1, 2, \cdots, K. \tag{12}$$

where $u_j$ and $v_k$ are the hazard indicator vectors. Further, the conditional probability distribution of the given compound hazards risk indicator $u$ has the expression of Eq. 13:

$$p_{v|u_j}(v_k|u_j) = \frac{p_{jk}}{\sum_{k=1}^{K} p_{jk}}, k = 1, 2, \cdots K. \tag{13}$$

For the two dimensional input risk indicator set (time and hazard level value) $A = \{(x_{1t}, x_{2t})|t = 1, 2, \cdots, T\}$ with diffusion function $\mu_A(u_j, v_k)$, the fuzzy relationship (vulnerability distribution) between $A$ and fuzzy output (economic losses indicator

$f_m$) $B = R_f$ can be estimated by membership function $\mu_B(f_m)$:

$$\mu_B(f_m) = \max_{\substack{u_j \in U \\ v_k \in V}} \{\min\{\mu_A(u_j, v_k), R_f\}\}, m = 1, 2, \cdots, M. \tag{14}$$

where the fuzzy relationship model $R_f = \{(r_{jkm})\}$ (Eq. 15) is given by the three-dimension information diffusion matrix $\mu(X_t; u_j, v_k, f_m)$.

$$r_{jkm} = \frac{\sum_{i=1}^{n} \mu(X^t{}_i; u_j, v_k, f_m)}{\max_{1 \leq m \leq M} \sum_{i=1}^{n} \mu(X^t{}_i; u_j, v_k, f_m)}. \tag{15}$$

Then the weighted value $f(u_j, v_k)$, represented as vulnerability distribution, is defined as follows.

$$f(u_j, v_k) = \frac{\sum_{m=1}^{M} \mu_B(f_m) \cdot f_m}{\sum_{m=1}^{M} \mu_B(f_m)}, j = 1, 2, \cdots, J; k = 1, 2, \cdots, K. \tag{16}$$

Based on the VFS-IDM risk assessment model, the dynamic compound hazards risk (Direct Economic Losses) is shown in Eq. 17 where the risk is quantified as the expected value of conditional probability distribution and vulnerability distribution. The proposed algorithm, which can be used to deal with incomplete information risk assessment, is given by Algorithm 2.

$$Risk_{u_j} = \sum_{k=1}^{K} p_{v|u_j}(v_k|u_j) \cdot f(u_j, v_k) \tag{17}$$

## 3   Case Study

This section uses the Typhoon-Rainstorm compound hazards that occurred in Shenzen, China, as an example to show how the proposed VFS-IEM-IDM model can be used to dynamically assess the risk of compound hazards. Shenzhen is located in the east bank of the Zhujiang River and is surrounded by Daya Bay and Dapeng Bay, where the climate is a subtropical maritime




---

**Algorithm 2** VFS-IDM Dynamic Risk Assessment Model of Compound Hazards

---

**Input:**

1: Sample set $D = \{X^t{}_i = (x^t{}_{1i}, x^t{}_{2i}, x^t{}_{3i}) | i = 1, 2, \cdots, n; t = 1, 2, \cdots, T\}$, where $x^t{}_{oi}, o = 1, 2, 3$ is the related factor of compound hazards (results given by Algorithm 1);

2: Universes of monitor space $S = \{(s_{ol}), l = 1, 2, \cdots, L | o = 1, 2, 3\}$, where the length $L$ varies from different universes;

3: Coefficients of diffusion function $H = (h_1, h_2, h_3)$.

**Output:**

Dynamic compound hazards risk.

4: Identification of the comprehensive value of compound hazards level by VFS-IEW;

5: **for** Sample index $i = 1$ to $n$, each $X^t{}_i$ **do**

6:     Based on the universes of monitor space, employing the normal diffusion function in Eq. 10 to construct information diffusion matrix of sample $D$;

7: **end for**

8: Estimate the joint and conditional probability distribution based on Eq. 12 and Eq. 13;

9: Determine the input-output sets and model the fuzzy relationship based on Eq.15, then estimate the vulnerability distribution by Eq.16;

10: The dynamic risk (Direct Economic Loss) of compound hazards can be quantified by Eq. 17.

---

and Typhoon-Rainstorms are undoubtedly the most frequently occurred hazards in Shenzhen. According to the collected data (see Table A1), from 1980 to 2016, on average, the directed economic losses of the Typhoon and Rainstorm hazards in Shenzhen exceeded 360 million RMB per year, the number of death was $3.4$ deaths annually and about $149,000$ people were affected (Zhou et al. (2017)). The assessment results of the Typhoon-Rainstorm dynamic risk are the basis to determine whether or not the early warning systems are worked and implemented effectively.

Since the compound hazards are charactered by three indicators, the variable fuzzy set dimension reduction model can be used to get more precise comprehensive hazard level. According to the Classification Standards of Rainstorm and Typhoon, this paper outlines the index classification criteria (shown in Table 1, Guided by TYPHOON ONLINE http://typhoon.nmc.cn/web.html and the explanation is shown in the Table A1) and the four types of Typhoon-Rainstorm hazard level.

**Table 1.** Classification standards of Typhoon-Rainstorm hazards.

| Indicators | Compound Hazards Level | | | |
| --- | --- | --- | --- | --- |
| | I | II | III | IV |
| Maximum Precipitation | (0,50) | (50,100) | (100,150) | (150,250) |
| Strong Wind Intensity | (8,10.8) | (10.8,17.2) | (17.2,23.6) | (23.6,30) |
| Transformed Location Number | (0,2) | (2,5) | (5,8) | (8,10) |

    Based on expert experiences and the relevant government documents, the classification results of Typhoon-Rainstorm hazard

level ($H$) in Shenzhen express type I as $H \in [1.5, 2)$, type II as $H \in [2, 2.7)$, type III as $H \in [2.7, 3.5)$, and type IV as $H \in$





[3.5, 4]. This paper uses the dimension reduction model VFS-IEM (Algorithm 1) to get the comprehensive value $H$ and transfers them into different hazard levels based on Typhoon-Rainstorm classification standards. According to the classification results shown in Table 1, the interval criterion matrix can be expressed as

$$I_{ab} = \begin{bmatrix} (0,50) & (50,100) & (100,150) & (150,250) \\ (8,10.8) & (10.8,17.2) & (17.2,23.6) & (23.6,30) \\ (0,2) & (2,5) & (5,8) & (8,10) \end{bmatrix} = ((a,b)_{rl}),$$


$$I_{cd} = \begin{bmatrix} (0,100) & (0,150) & (50,250) & (100,250) \\ (8,17.2) & (8,23.6) & (10.8,30) & (17.2,30) \\ (0,5) & (0,8) & (2,10) & (5,10) \end{bmatrix} = ((c,d)_{rl}),$$

and the balance boundaries matrix $M$ is defined in Eq. 18

$$M = \begin{bmatrix} 0 & 66.7 & 133.3 & 250 \\ 8 & 12.9 & 21.5 & 30 \\ 0 & 3 & 7 & 10 \end{bmatrix} = (M_{rl}). \tag{18}$$

Then the relative membership degree matrix can be calculated by Eqs. 5, 6 and 7 respectively.

Taking sample point $(MP = 33.4, SWI = 18, TL = 9)$ for example, the relative membership degree matrix is expressed by Eq. 19 where the matrix value represents the probability of each indicator belonging to the different compound hazards level.

$$\mu_A(u) = \begin{bmatrix} 0.666 & 0.334 & 0.000 & 0.000 \\ 0.000 & 0.438 & 0.593 & 0.063 \\ 0.000 & 0.000 & 0.333 & 0.667 \end{bmatrix}. \tag{19}$$

To get the comprehensive hazard level, the information entropy method can be used to get the weight of each indicator $\omega$, which implies that the Maximum Precipitation and Location play the main role in determining the Typhoon-Rainstorm hazards

level.

$$\omega = \begin{pmatrix} 0.43 & 0.19 & 0.39 \end{pmatrix}. \tag{20}$$

Then by Algorithm 1, the comprehensive value of the Typhoon-Rainstorm hazards level $(MP = 33.4, SWI = 18, TL = 9)$ is $H = 2.75$ when the hyper-parameter $\alpha = \beta = 1$, and $H = 2.18$ when the hyper-parameter $\alpha = \beta = 2$. To be more general, this paper takes the average of $H = 2.75$ and $H = 2.18$ to obtain the final compound hazards level, i.e., $H = 2.47$, Type II.

The results of other Typhoon-Rainstorm comprehensive hazard levels can be found in Appendix (see Table B1).




Table 2: Transformed Typhoon-Rainstorm hazard data sets in Shenzhen.

| Time | Transformed Time ($T$) | Comprehensive Hazard Level ($H$) | Direct Economic Loss ($L$) |
|---|---|---|---|
| 20090627 | 176 | 2.72 | 0.3819 |
| 0719 | 198 | 3 | 1.352 |
| 0915 | 254 | 3.74 | 1.3750 |
| 20100724 | 203 | 2.32 | 0.2571 |
| 0912 | 251 | 2.49 | 0.4450 |
| 0922 | 261 | 2.74 | 0.9831 |
| 20110624 | 173 | 1.93 | 0.0765 |
| 0930 | 269 | 2.72 | 0.4013 |
| 20120630 | 179 | 2.31 | 0.2895 |
| 0724 | 203 | 3.95 | 2.48 |
| 0817 | 226 | 2.56 | 0.7648 |
| 20130615 | 164 | 1.94 | 0.1527 |
| 0702 | 181 | 1.99 | 0.1894 |
| 0802 | 211 | 1.53 | 0.0452 |
| 0814 | 223 | 2.13 | 0.1423 |
| 0922 | 261 | 3.06 | 1.2351 |
| 20140718 | 197 | 1.83 | 0.0841 |
| 0916 | 255 | 2.48 | 0.7682 |
| 0823 | 232 | 2.92 | 0.7410 |
| 1004 | 273 | 2.96 | 0.8352 |
| 20160802 | 211 | 3.68 | 2.1521 |
| 0818 | 227 | 1.88 | 0.0251 |
| 1018 | 287 | 2.28 | 0.2362 |
| 1021 | 290 | 3.11 | 0.9341 |
| 20170612 | 161 | 3.67 | 2.058 |
| 0723 | 202 | 2.11 | 0.2461 |
| 0823 | 232 | 2.46 | 1.31 |
| 0827 | 236 | 3.2 | 1.613 |
| 0903 | 242 | 3.03 | 1.8872 |
| 1016 | 285 | 2.48 | 0.5902 |
| 20180606 | 155 | 2.47 | 0.6952 |




| 0718 | 197 | 1.58 | 0.0267 |
|---|---|---|---|
| 0811 | 220 | 2.45 | 0.5241 |
| 0916 | 255 | 3.93 | 2.226 |
| 20190703 | 182 | 1.49 | 0.0528 |
| 0811 | 210 | 3.02 | 0.8182 |
| 0824 | 233 | 2.9 | 0.8391 |
| 0902 | 241 | 1.8 | 0.0725 |

From Table 2, the sample observations on direct economic loss $L$ (Billions) over each comprehensive compound hazards level $H$ are written as

$$Sample = \{(t_1, d_1, l_1), \ldots, (t_i, d_i, l_i), \ldots, (t_{38}, d_{38}, l_{38})\} = \{(172, 2.72, 0.3819), \ldots, (241, 1.8, 0.0725)\}.$$

where $t_i$, $d_i$ represents the time dimension of the Typhoon-Rainstorm hazard and the comprehensive value of the hazards level respectively, and $l_i$ is the direct economic losses caused by the Typhoon-Rainstorm hazards. Then the diffusion coefficients can be calculated by Eq. 11, written as

$$\begin{cases} h_t = 2.6581 \cdot (290 - 155)/(38 - 1) = 10 \\ h_d = 2.6581 \cdot (3.95 - 1.37)/(38 - 1) = 0.19 \\ h_l = 2.6581 \cdot (2.48 - 0.0251)/(38 - 1) = 0.1764 \end{cases}.$$

Algorithm. 2 outlines how to use the information diffusion method to estimate the conditional probability and vulnerability
distribution of the Typhoon-Rainstorm hazards. Then by the 2-dimensional normal diffusion estimator, the joint probability density function $P$ (Eq. 21) and conditional probability function $P_{con}$ (Eq. 22) can be evaluated. In this paper, we denote the monitor space $T = (t = 164, t = 194, t = 224, t = 254, t = 284)$ as months $(June, July, August, September, October)$ and $H = (d = 1.8, d = 2.4, d = 3.0, d = 3.6)$ as comprehensive hazards levels $(I, II, III, IV)$.

$$P = \begin{matrix} & \begin{matrix} I & II & III & IV \end{matrix} \\ \begin{matrix} June \\ July \\ August \\ September \\ October \end{matrix} & \begin{bmatrix} 0.059 & 0.046 & 0.007 & 0.036 \\ 0.076 & 0.052 & 0.051 & 0.014 \\ 0.063 & 0.116 & 0.090 & 0.019 \\ 0.019 & 0.086 & 0.087 & 0.041 \\ 0.002 & 0.073 & 0.060 & 0.002 \end{bmatrix} \end{matrix}, \tag{21}$$





$$
P_{con} = \begin{array}{c} \\ June \\ July \\ August \\ September \\ October \end{array}
\begin{array}{cccc} I & II & III & IV \\ \\ \end{array}
\left[\begin{array}{cccc}
0.398 & 0.311 & 0.049 & 0.243 \\
0.393 & 0.268 & 0.266 & 0.073 \\
0.218 & 0.402 & 0.312 & 0.068 \\
0.080 & 0.370 & 0.373 & 0.177 \\
0.012 & 0.539 & 0.437 & 0.012
\end{array}\right]. \tag{22}
$$

From the results above, it can be seen that the Typhoon-Rainstorm hazard level of III occur more frequently and they are most likely to occur in August and September.

The vulnerability distribution $f(x)$ between the comprehensive value $H$ and the direct economic losses $L$ over the time dimension $T$ can be calculated by the 3-dimension diffusion estimator. The fuzzy relationship which takes time dimension $T$, hazards level $H$ as input and the loss $L$ as the output can be denoted as matrix $R_f$.

$$
R_f = \left(\begin{array}{c}
\begin{array}{c}
l=0.1 \quad l=0.4 \quad l=0.7 \quad l=1.0 \quad l=1.3 \quad l=1.6 \quad l=1.9 \quad l=2.2
\end{array} \\
t=164 \begin{array}{c} d=1.8 \\ d=2.4 \\ d=3.0 \\ d=3.6 \end{array}
\left[\begin{array}{cccccccc}
0.80 & 0.43 & 0.01 & 0.00 & 0.00 & 0.00 & 0.00 & 0.00 \\
0.15 & 0.42 & 0.57 & 0.15 & 0.00 & 0.00 & 0.00 & 0.00 \\
0.00 & 0.05 & 0.13 & 0.04 & 0.00 & 0.00 & 0.00 & 0.00 \\
0.00 & 0.00 & 0.00 & 0.00 & 0.00 & 0.08 & 1.00 & 1.00
\end{array}\right] \\
t=194 \begin{array}{c} d=1.8 \\ d=2.4 \\ d=3.0 \\ d=3.6 \end{array}
\left[\begin{array}{cccccccc}
1.00 & 0.49 & 0.01 & 0.00 & 0.00 & 0.00 & 0.00 & 0.00 \\
0.45 & 0.82 & 0.10 & 0.01 & 0.01 & 0.00 & 0.00 & 0.00 \\
0.00 & 0.04 & 0.22 & 0.28 & 1.00 & 0.94 & 0.02 & 0.00 \\
0.00 & 0.00 & 0.00 & 0.00 & 0.01 & 0.00 & 0.13 & 0.43
\end{array}\right] \\
\vdots \\
t=284 \begin{array}{c} d=1.8 \\ d=2.4 \\ d=3.0 \\ d=3.6 \end{array}
\left[\begin{array}{cccccccc}
0.02 & 0.03 & 0.00 & 0.00 & 0.00 & 0.00 & 0.00 & 0.00 \\
0.35 & 1.00 & 0.61 & 0.08 & 0.00 & 0.00 & 0.00 & 0.00 \\
0.02 & 0.15 & 0.56 & 1.00 & 0.19 & 0.03 & 0.00 & 0.00 \\
0.00 & 0.00 & 0.01 & 0.03 & 0.01 & 0.00 & 0.00 & 0.00
\end{array}\right]
\end{array}\right).
$$





The discrete vulnerability distribution in terms of the direct economic loss is evaluated by Eq. 16 and the results are shown in Eq. 23. It can be seen that Shenzhen, most of the economic losses caused by the Typhoon-Rainstorm hazards is concentrated in August and September.

$$f(x;t,h) = \begin{array}{c} \\ June \\ July \\ August \\ September \\ October \end{array} \begin{array}{cccc} I & II & III & IV \\ \left[ \begin{array}{cccc} 0.20 & 0.02 & 0.00 & 0.00 \\ 0.24 & 0.04 & 0.00 & 0.00 \\ 0.15 & 1.13 & 1.67 & 1.90 \\ 0.05 & 0.55 & 2.67 & 2.62 \\ 0.01 & 0.02 & 0.00 & 0.00 \end{array} \right] \end{array}. \tag{23}$$

Dynamic compound hazards risks can be quantified as the expected value of hazards influence and the result is shown as Eq. 24 where the elements of vector denotes the estimated economic losses caused by the Typhoon-Rainstorm hazards in different months.

$$Risk = \left( \begin{array}{ccccc} 0.08582 & 0.10504 & 1.1372 & 1.66715 & 0.0109 \end{array} \right). \tag{24}$$

## 4 Discussion

Dimension reduction model VFS-IEW presents the comprehensive value of compound hazards level, but the relationship between hazards level and the indicators are not clear. To find more information from the results of VFS-IEM, this paper has built a predictive model to shield the light on compound hazards relationship and predict the Typhoon-Rainstorm hazards level. Since the compound hazards level is an ordinal data (monotone trend and proportional odds), the cumulative logistic model (shown in Eq. 25) can be used to predict the compound hazards level. The probabilities of different order categories given by cumulative logistic model are

$$P(Y \le j \mid x) = \pi_1(x) + \cdots + \pi_j(x), \quad j = 1,\ldots,J.$$

According to the research by Alan (1980), the cumulative logistic model is defined as

$$\text{logit}[P(Y \le j \mid x)] = \log \frac{P(Y \le j \mid x)}{1 - P(Y \le j \mid x)} = \text{logit}[P(Y \le j \mid x)] = \alpha_j + \boldsymbol{\beta}^T \boldsymbol{x}, \quad j = 1,\ldots,J-1. \tag{25}$$

The Typhoon-Rainstorm hazard level prediction problem can be solved by using the VAGM package (Thomas (2010)) and the result is given by

$$\text{logit}[P(Y \le j \mid x)] = 5.07(7.32, 11.15) - 0.12MP - 0.66SWI - 0.91TLN, \tag{26}$$

where the different intercepts denote the different main-effects of hazard levels compared to the reference category, i.e., hazard level IV. The rationality of this model is judged by LR-test (p-value<0.001) and the predictive performance $R^2 = 0.898$ which shows that the model is well fitted and can be used as the compound hazards prediction.



One advantage using the information diffusion technique to assess the risk of an compound hazards is that it does not need to know (1) the distribution type of the population from which given samples are drawn, (2) the function form of the causal relationship, which are constructed by the joint probability distribution and the vulnerability distribution. Moreover, researchers

have done simulation study on IDM and demonstrate the benefit of information distribution for probability estimation (Huang (2000); Li et al. (2012)) by minimizing the mean integrated square error (Kullback-Leibler divergence error) between the estimator and the true density. The performance of this non-parametric estimation procedure is studied well by Huang (2000) which shows the work efficiency is about 35% higher than histogram estimator (HE) and the performance is improved to reduce the error by 23.2% when data sets are incomplete. Therefore, the assessed compound hazards risk is more reliable and

accurate.

For the dynamic risk assessment of Typhoon-Rainstorm hazards, this paper provides extensive assessment results. From the dimension reduction model VFS-IEM, this paper shows that the probability of the occurrences of type II and III hazard levels is highest in Shenzhen. The emergency management department should prepare more effective emergency plans in advance to reduce the occurrences of the secondary hazards. From the dynamic risk assessment model VFS-IEM-IDM , it can be found

that the hazards occurrence probability of different hazard levels is different and the type hazards of II and III hazards level are most likely to occur in August and September. Also, considering the occurrence of different hazard level for each months, the probability of hazard level I occurring in June and July is the highest, and the hazard level II mostly occurs in August and October, and the type III hazard level is most likely to occur in September. From the perspective of hazard losses, the different direct economic losses caused by the Typhoon-Rainstorms of the same hazard level in each month indicates that the impacts

of the Typhoon-Rainstorm hazards on the economy are not the same. Besides, for the same month, the influence of economic loss decreases gradually when the compound hazards level rises. This indicates that the capacity of Typhoon-Rainstorm hazard resistance in Shenzhen is reliable, and the ability to copy with the sudden compound hazards are relatively strong under the existing emergence system. The dynamic compound hazards risk of Typhoon-Rainstorm hazards in Shenzhen shows that the risk value of this compound hazards in each month is different and the highest risk value appears in August and September. On

average, the occurrence of Typhoon-Rainstorm hazards brought Shenzhen 114 million RMB and 167 million RMB losses in these two months respectively, which is in line with the actual situation.

## 5   Conclusions

Risk assessment is an important step in emergency management, but little research discusses the uncertainties of compound hazards evaluation and considers dynamic risk assessments when the data sets are incomplete. In this paper, we first present

the definition of dynamic compound hazards risk, and then Variable Fuzzy Set (VFS) theory is employed to evaluate the relative membership degree, and Information Entropy Method (IEM) is applied to obtain the weights of criteria indicators for compound hazards level evaluation. Based on the results obtained by VFS-IEM, we apply the information diffusion method (IDM) to estimate the conditional probability distribution and vulnerability distribution with the hazard occurrence time and the corresponding losses. Then the dynamic risk is assessed using fuzzy probabilistic risk to improve the accuracy of risk



assessment. The innovations of this paper are: (i) Based on the definition of compound hazards risk, we take time dimension into consideration to introduce the concept of dynamic compound hazards risk. (ii) Considering that compound hazards have different measurement indicators for the comprehensive evaluation, a hybrid model of Variable Fuzzy Sets and the Information Entropy Method has been proposed to improve the accuracy of compound hazards level evaluation. (iii) According to the concept of dynamic compound hazards risk, we apply Information Diffusion Method to estimate the conditional probability

distribution and the vulnerability distribution using the comprehensive hazard levels, hazards occurrence time and the losses of compound hazards. The proposed model VFS-IEM-IDM can be used to deal with the problem of incomplete and limited information in dynamic risk assessment. (iv) By evaluating the expected value of the conditional probability distribution and the vulnerability distribution, we quantify the Typhoon-Rainstorm dynamic risk which shows that the occurrence of Typhoon-Rainstorm hazards brings Shenzhen 114 million RMB and 167 million RMB losses in August and September respectively.

These risk assessment results are in line with the actual situation and may be used to guide the emergency management in Shenzhen, which also shows the potential of VFS-IEM-IDM being applied to the compound hazards in general.

*Code and data availability.*  The data and code used in the study are available at https://github.com/GongWenwuu/VFS-IEM-IDM.git.

*Acknowledgements.*  This work was mainly supported by the National Key Research and Development Program of China (grant nos.
2019YFC0810705 and 2018YFC0807000), and National Natural Science Foundation of China under Grant 71771113. The authors would

like to acknowledge Shuanghua Yang and Manyu Meng of the Southern University of Science and Technology for providing the useful information.

**Author contributions.** GWW and YLL conceived the research framework and developed the methodology. GWW was responsible for the code compilation, data analysis, graphic visualization. GWW and JJ had done the first draft writing. YLL managed the implementation of research activities and revised the manuscript. All authors discussed the results and contributed

to the final version of the paper.

**Competing interests.** The authors declare that they have no conflict of interest.

**Special issue statement.** This article is part of the special issue 'Advances in flood forecasting and early warning'.






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





## Appendix A: Data Source

For the Typhoon-Rainstorm dynamic compound hazards risk assessment, the useful data sets, collected from Meteorological Bureau of ShenZhen Municipality (http://weather.sz.gov.cn/qixiangfuwu/qihoufuwu/qihouguanceyupinggu/nianduqihougongbao/) and TYPHOON ONLINE (http://typhoon.nmc.cn/web.html), have been sorted out in Table A1. In this table, MP denotes as
Maximum Precipitation, SWI denotes as Strong Wind Intensity, DEL denotes as Direct Economic. The Transformed Location Number (TLN) is also denoted as the landing location by using expertise knowledge.

Table A1: Data sets of Typhoon-Rainstorm hazards in Shenzhen.

| Time | MP ($mm$) | SWI ($m/s$) | Landing Location | Transformed Location | DEL (Billion) |
|---|---|---|---|---|---|
| 20090627 | 67.3 | 16.8 | Huizhou | 8.5 | 0.3819 |
| 0719 | 80 | 27.3 | Shenzhen | 10 | 1.152 |
| 0915 | 127.9 | 28 | Taibei | 6 | 1.075 |
| 20100724 | 54.3 | 16.2 | Zhanjiang | 6.5 | 0.2571 |
| 0912 | 62.4 | 13.7 | Quanzhou | 3 | 0.345 |
| 0922 | 51.9 | 15.8 | Heyuan | 7 | 0.2983 |
| 20110624 | 41.7 | 14 | Yangjiang | 4.5 | 0.0765 |
| 0930 | 117.3 | 15.2 | Wenchang | 2.5 | 0.8243 |
| 20120630 | 33.6 | 16.8 | Zhuhai | 6.5 | 0.6873 |
| 0724 | 152.3 | 24.9 | Taishan | 7 | 2.241 |
| 0817 | 66.1 | 13.5 | Zhanjiang | 3 | 0.9153 |
| 20130615 | 36.5 | 12.4 | Wenchang | 4 | 0.3621 |
| 0702 | 38.6 | 13.9 | Zhanjiang | 3 | 0.2561 |
| 0802 | 40.7 | 14.7 | Wenchang | 3 | 0.0851 |
| 0814 | 47.8 | 14.2 | Yangxi | 3 | 0.6413 |
| 0922 | 72.4 | 21.6 | Shanwei | 8.5 | 1.152 |
| 20140718 | 34.6 | 14.7 | Wenchang | 2.5 | 0.0841 |
| 0916 | 73.5 | 18.9 | Xuwen | 2.5 | 0.9641 |
| 0823 | 69.4 | 13.6 | Shanwei | 10 | 1.041 |
| 1004 | 108.5 | 13.5 | Zhanjiang | 5.5 | 0.9631 |
| 20160802 | 166 | 19.2 | Shenzhen | 10 | 2.31 |
| 0818 | 45.5 | 11.1 | Zhanjiang | 5.5 | 0.0314 |
| 1018 | 117.6 | 12.3 | Wanning | 1.5 | 0.421 |
| 1021 | 83.7 | 18.8 | Shanwei | 7.5 | 0.8721 |



| | | | | | |
|---|---|---|---|---|---|
| 20170612 | 161.8 | 23 | Shenzhen | 10 | 2.109 |
| 0723 | 33.4 | 18 | Xianggang | 9 | 0.5315 |
| 0823 | 56.3 | 23.9 | Zhuhai | 8.5 | 1.328 |
| 0827 | 114.5 | 17.5 | Jiangmen | 8.5 | 1.741 |
| 0903 | 82.4 | 14.4 | Shanwei | 7.5 | 0.9631 |
| 1016 | 40 | 20.3 | Zhanjiang | 7.5 | 0.7341 |
| 20180606 | 97.2 | 10.8 | Xuwen | 8.5 | 0.9267 |
| 0718 | 50.7 | 11.1 | Wanning | 1.5 | 0.0267 |
| 0811 | 45.3 | 10.8 | Yangjiang | 7 | 0.5241 |
| 0916 | 173.5 | 30 | Taishan | 7.5 | 2.361 |
| 20190703 | 48.8 | 11 | Wanning | 1.5 | 0.0672 |
| 0811 | 178.5 | 14.1 | Wenchang | 5.5 | 0.9561 |
| 0824 | 97.6 | 12.7 | Zhangzhou | 6 | 0.5931 |
| 0902 | 86.9 | 11.3 | Wanning | 1 | 0.0751 |




## Appendix B: Comprehensive Compound Hazards Level

Based on the Dimension Reduction Model VFS-IEM, this paper takes the average of $\alpha = \beta = 1$ and $\alpha = \beta = 2$ to denote
the final Typhoon-Rainstorm hazards level. The following Table B1 has shown that the whole results of comprehensive degree
value.

Table B1: Comprehensive compound hazards level in ShenZhen

| Time | $\alpha = \beta = 1$ | $\alpha = \beta = 2$ | Average Level (D) | Comprehensive Level |
|---|---|---|---|---|
| 20090627 | 3.07 | 2.36 | 2.72 | III |
| 0719 | 3.34 | 2.65 | 3.00 | III |
| 0915 | 3.93 | 3.55 | 3.74 | IV |
| 20100724 | 2.67 | 1.96 | 2.32 | II |
| 0912 | 2.68 | 2.29 | 2.49 | III |
| 0922 | 3.02 | 2.45 | 2.74 | III |
| 20110624 | 2.12 | 1.73 | 1.93 | I |
| 0930 | 2.87 | 2.57 | 2.72 | III |
| 20120630 | 2.66 | 1.95 | 2.31 | II |
| 0724 | 3.97 | 3.93 | 3.95 | IV |
| 0817 | 2.8 | 2.32 | 2.56 | II |
| 20130615 | 2.08 | 1.79 | 1.94 | I |
| 0702 | 2.28 | 1.7 | 1.99 | I |
| 0802 | 1.65 | 1.4 | 1.53 | I |
| 0814 | 2.22 | 2.03 | 2.13 | II |
| 0922 | 3.44 | 2.67 | 3.06 | III |
| 20140718 | 1.93 | 1.73 | 1.83 | I |
| 0916 | 2.65 | 2.3 | 2.48 | II |
| 0823 | 3.19 | 2.64 | 2.92 | III |
| 1004 | 3 | 2.91 | 2.96 | III |
| 20160802 | 3.66 | 3.69 | 3.68 | IV |
| 0818 | 1.96 | 1.8 | 1.88 | I |
| 1018 | 2.52 | 2.03 | 2.28 | II |
| 1021 | 33.1 | 2.91 | 3.11 | III |
| 20170612 | 3.69 | 3.83 | 3.76 | IV |
| 0723 | 2.52 | 1.7 | 2.11 | II |



| | | | | |
|---|---|---|---|---|
| 0823 | 2.89 | 2.03 | 2.46 | II |
| 0827 | 3.35 | 3.04 | 3.2 | III |
| 0903 | 3.22 | 2.83 | 3.03 | III |
| 1016 | 2.95 | 2 | 2.48 | II |
| 20180606 | 2.75 | 2.18 | 2.17 | II |
| 0718 | 1.57 | 1.45 | 1.51 | I |
| 0811 | 2.72 | 2.17 | 2.45 | II |
| 0916 | 3.87 | 3.98 | 3.93 | IV |
| 20190703 | 1.52 | 1.48 | 1.5 | I |
| 0811 | 3.25 | 2.79 | 3.02 | III |
| 0824 | 2.96 | 2.83 | 2.9 | III |
| 0902 | 1.93 | 1.67 | 1.8 | I |