# Peer review of "Dynamic Risk Assessment of Compound Hazards Based on VFS-IEM-IDM: A Case Study of Typhoon-rainstorm Hazards in Shenzhen, China"

_Natural Hazards and Earth System Sciences, 2021_

## Author Response (AR1)

**Responses letter**

We are very grateful to the reviewers for his/her constructive suggestions for this manuscript, which is a great help and guidance for this study and our future research. Here are our responses to the comments from two reviewers and the details of how we made the changes in our manuscript.

**Responses to the comments from the anonymous referee #1**

1. "The matrix in Line 237 must be reformatted (see attached file)".

   Thanks to the reviewer for pointing out the wrong expression of formula. We have reformatted the matrix in the Manuscript as follows.

$$
P_{con} = \begin{pmatrix}
 & I & II & III & IV \\
June & 0.398 & 0.311 & 0.049 & 0.243 \\
July & 0.393 & 0.268 & 0.266 & 0.073 \\
August & 0.218 & 0.402 & 0.312 & 0.0689 \\
September & 0.080 & 0.370 & 0.373 & 0.177 \\
October & 0.012 & 0.539 & 0.437 & 0.012
\end{pmatrix}
$$

$$
R_f = \begin{pmatrix}
t=164 & \begin{array}{c} d=1.8 \\ d=2.4 \\ d=3.0 \\ d=3.6 \end{array} & \left[\begin{array}{cccccccc}
 & l=0.1 & 0.4 & 0.7 & 1.0 & l=1.3 & 1.6 & 1.9 & 2.2 \\
0.80 & 0.43 & 0.01 & 0.00 & 0.00 & 0.00 & 0.00 & 0.00 \\
0.15 & 0.42 & 0.57 & 0.15 & 0.00 & 0.00 & 0.00 & 0.00 \\
0.00 & 0.05 & 0.13 & 0.04 & 0.00 & 0.00 & 0.00 & 0.00 \\
0.00 & 0.00 & 0.00 & 0.00 & 0.00 & 0.08 & 1.00 & 1.00
\end{array}\right] \\
t=194 & \begin{array}{c} d=1.8 \\ d=2.4 \\ d=3.0 \\ d=3.6 \end{array} & \left[\begin{array}{cccccccc}
1.00 & 0.49 & 0.01 & 0.00 & 0.00 & 0.00 & 0.00 & 0.00 \\
0.45 & 0.82 & 0.10 & 0.01 & 0.01 & 0.00 & 0.00 & 0.00 \\
0.00 & 0.04 & 0.22 & 0.28 & 1.00 & 0.94 & 0.02 & 0.00 \\
0.00 & 0.00 & 0.00 & 0.00 & 0.01 & 0.00 & 0.13 & 0.43
\end{array}\right] \\
 & & \vdots \\
t=284 & \begin{array}{c} d=1.8 \\ d=2.4 \\ d=3.0 \\ d=3.6 \end{array} & \left[\begin{array}{cccccccc}
0.02 & 0.03 & 0.00 & 0.00 & 0.00 & 0.00 & 0.00 & 0.00 \\
0.35 & 1.00 & 0.61 & 0.08 & 0.00 & 0.00 & 0.00 & 0.00 \\
0.02 & 0.15 & 0.56 & 1.00 & 0.19 & 0.03 & 0.00 & 0.00 \\
0.00 & 0.00 & 0.01 & 0.03 & 0.01 & 0.00 & 0.00 & 0.00
\end{array}\right]
\end{pmatrix}
$$

2. "The grammar in the article needs to be improved. For example, Line 30: There are many works discussing the multi-hazard risk assessment and Choi et al. (2021) had reviewed the relevant literature. ==> There are many works discussing the multi-hazard risk assessment which have been reviewed by Choi et al. (2021)".

   Thanks to the reviewer for the careful grammar checking on this manuscript. We have improved the grammar writing and the detailed can be seen in our Manuscript.

30  There are many works discussing the multi-hazard risk assessment  which have been reviewed the relevant literature Choi et al. (2021). Furthermore, Wang et al. (2020) clarified the relationship between hazards in multi-

**Responses to the comments from the anonymous referee #2**

1. "Lines 74-81, if I understand it correctly based on the description, the main contribution of the paper is reflected in the first point, and the others are the improvement and verification of this method. Therefore, integrating this part with the last paragraph of introduction is recommended. At the end of the introduction, it is recommended to clearly point out the innovations of the paper and the main contributions of the authors. Technological innovation?"

Thanks for your constructive suggestions for this manuscript. We did not clearly point out the innovations of the paper and the main contributions of the authors. As the reviewer said, some technological innovation and the application of proposed model have been concluded at the end of the introduction.

The main contributions of this paper are summarized as following two-folds.

- 1)  For technological innovation, we propose a hybrid model, named as Variable Fuzzy Set and Information Diffusion Method (VFS-IEM-IDM), to  assess compound hazards risk

-  dynamically. Furthermore, we simplify the calculation procedures of relative membership degree to improve the efficiency and accuracy of compound hazards level evaluation

-  2) To examine the efficacy of the proposed model VFS-IEM-IDM, a case study of the Typhoon-Rainstorm hazards that occurred in Shenzhen, China is presented.

2. "Table 1. Classification standards of Typhoon-Rainstorm hazards. What is the reference for this standard? The precipitation in this table is the daily maximum precipitation or total precipitation? The strong wind here represents the maximum wind speed or extreme wind speed? Please define and explain them in detail."

Thanks to the reviewer for pointing out the lack of explanation of the Classification standards of Typhoon-Rainstorm hazards in the original manuscript. We have look through some documents given by Shenzhen Climate Bulletin and tried our best to define and explain the classification criteria in detail.

Since the Typhoon-Rainstorm compound hazards are characteried by three indicators  (Maximum Daily Precipitation, Extreme Wind Intensity, Transformed Location Number), the variable fuzzy set dimension reduction model can be used to get more precise comprehensive hazard level.  This paper has outlined the index classification criteria (shown in Table 1and the four types of ), which is guided by Shenzhen Climate Bulletin (http: //weather.sz.gov.cn/qixiangfuwu/qihoufuwu/). We also classify the Typhoon-Rainstorm compound hazard level into four types, and which is related to our final results.

3. "The monthly differences of different types of disasters may be closely related to the frequency of typhoons and the intensity of typhoons. What are the considerations in this paper?"

Thanks to the reviewer for the question on the key process of dynamic risk assessment. As the reviewer said the monthly differences of different types of disasters may be closely related to the frequency of typhoons and the intensity of typhoons, we have taken the time dimension into consideration to deal with the time attribute so as to assess the risk dynamically.

4. "The data in the "Table A1"is the precipitation and strong wind data during the period affected by the typhoon rainstorms in Shenzhen, China. The source is the website of Shenzhen Meteorological Bureau. But there are big problems with the data in the table."

Thanks to the reviewer for the careful review of this manuscript. We have checked the data source more carefully in the Supporting Information and we have solved the data is inconsistent. The detail can be seen the Manuscript.

as Direct Economic  Loss, and the Transformed Location Number (TLN)  denotes as the Typhoon Landing Location which is determined by radio distance transform using expertise knowledge.

Table A1: Data sets of Typhoon-Rainstorm hazards in Shenzhen.

| Hazards Number | Impact Time |  MDP (mm) |  EWI (m/s) | Landing Location |  TLN | DEL |
|---|---|---|---|---|---|---|
|  0904 | 0627 | 67.3 | 16.8 | Huizhou | 8.5 | 0 |
| 0906 | 0719 | 80 | 27.3 | Shenzhen | 10 | 1 |
| 0915 | 0912 | 127.9 | 28 | Taibei | 6 | 1 |
|  1003 | 0724 | 31.3 | 16.2 | Zhanjiang | 6.5 | 0 |
|  1804 | 0606 | 97.2 |  8.8 | Xuwen | 8.5 | 0.9 |
| 1809 | 0718 | 50.7 | 11.1 | Wanning | 1.5 | 0.0 |
| 1816 | 0811 | 45.3 | 10.8 | Yangjiang | 7 | 0.5 |
| 1822 | 0916 | 173.5 | 30 | Taishan | 7.5 | 2.3 |
|  1904 | 0703 | 48.8 | 11 | Wanning | 1.5 | 0.0 |
| 1907 | 0811 |  99.1 | 14.1 | Wenchang | 5.5 | 0.9 |
| 1911 | 0824 |  49.4 | 12.7 | Zhangzhou | 6 | 0.5 |
| 1914 | 0902 |  52.2 | 11.3 | Wanning | 1 | 0.0 |

5. "Line 101, please provide references as evidence. Line 194, please provide references or related websites."

Thanks for your suggestions. The references have been added in the Manuscript as follows.

Though these four types of risks have been investigated by many researchers, there is little research on dynamic compound hazards risk (Huang et al. (2018)). In this paper, we give a definition of dynamic compound hazards risk and illustrate how to assess this kind of risks.

Since the Typhoon-Rainstorm compound hazards are characteried by three indicators  (Maximum Daily Precipitation, Extreme Wind Intensity, Transformed Location Number), the variable fuzzy set dimension reduction model can be used to get more precise comprehensive hazard level.  This paper has outlined the index classification criteria (shown in Table 1

195 ), which is guided by Shenzhen Climate Bulletin (http://weather.sz.gov.cn/qixiangfuwu/qihoufuwu/). We also classify the Typhoon-Rainstorm  compound hazard level into four types, and which is related to our final results.

6. "The limitations of this study and future plans are suggested to be added in the conclusion." Thanks for your suggestions on conclusion. We have added the limitations of this study and future work in the conclusion part.

guide the emergency management in Shenzhen  Dynamic risk assessment is a relatively new topic and we have proposed a hybrid model to assess the compound hazards. risk, but there are somewhere need to be improved. On one hand, the weight calculation of different types of hazards indicators

15 is subjective and the results of vulnerability curve have not taken the changes in the internal attributes of the affected area into consideration. On the other hand, there are also some subjective issues regarding how to process the data sets, so maybe we can consider adopting a more scientific method to process the original data to obtain more scientific conclusions.

---

## Author Response (AR2)

**Responses letter**

We are very grateful to the reviewer for his/her constructive suggestions for this manuscript, which is a great help and guidance for this study and our future research. Here are our responses to the comments from the reviewer and the details of how we made the changes in our manuscript.

**Responses to the comments from the anonymous referee**

1. In the introduction part, the definitions of VFS, IEM, IDM were poorly presented, I suggest adding more clear descriptions of them.

   Thanks to the reviewer for pointing out the poorly explanation of some key definitions. We have re-written the introduction part and added some preliminaries in Section 2.

   ## 2 Preliminaries

   ### 2.1 Basic concepts

   *Variable fuzzy set* is used to express the fuzzy effect of the hazard drivers by relative membership degree (RMD) functions, and then the compound effects between different drivers can be modeled. This method provides an enhanced implementation of the compound hazards level evaluation process and can reflect the coupled characteristics of compound hazards. *Information entropy* is based on the entropy coefficient calculation process, which is used to measure the importance of the individual hazard drivers and determine the weight of different drivers. *Information diffusion* is a function learning method with high estimation accuracy from a small data set, which makes full use of the diffusion information given by the data samples to estimate the probability density of the data samples or the relationship between the data samples without the knowledge of the distribution from which the data samples were drawn. This method is applied to estimate the probability distribution $p$ (hazard potential) of the occurrence of hazards, and the causal relationship $f$ (hazard vulnerability).

2. "The advantages and usage of the combined model should be declared more reliable.

Thanks to the reviewer for the comments on this manuscript. As the reviewer suggested, some technological innovations have been concluded at the end of the introduction. We have also presented the workflow of our proposed model which shows a detailed procedure and gives useful results of case study. Furthermore, we have illustrated the model efficient via a predict model and discussed the superiority of the normal diffusion estimator in the Discussion part.

[Figure]

3. In the part of case study, figures might be more readable than formulas and code.

Thanks for your constructive suggestions for this manuscript. We gave a more detailed introduction in the case study part by setting up multi-level headings. Furthermore, we hope that the major revision corresponded with the steps in the workflow proposed in Section 2, which can deepen the understanding of the proposed model.

---

## Author Response (AR3)

**Responses letter**

We are very grateful to the reviewers for his/her constructive suggestions for this manuscript, which is a great help and guidance for this study and our future research. Here are our replies for these comments, the detail can be seen in our Revised Manuscript.

**Responses letter for Anonymous Referee #2**

1. Figure 1 shows the workflow of the model, but both clarity and aesthetics need to be further optimized. Whether some special connecting lines have special meaning, it is suggested to add notes.

Thanks to reviewer for the careful review of this manuscript. We have updated a new figure and a more detailed explanation of proposed model framework in the Manuscript. There is no special meaning of the connecting lines, and we use the black rectangle to denote different calculation modules and use the blue one to represent the results obtained by the VFS-IEM-IDM model.

[Figure]

**Figure 1.** Workflow  of the VFS-IEM-IDM dynamic compound hazards risk assessment model for Typhoon-rainstorm hazards. Based on the historical records of  Typhoon-rainstorm hazards, our proposal provides two parts procedures: firstly, an enhanced implementation of the compound hazards level evaluation is proposed to assess the Typhoon-rainstorm hazards level; and then  the probability distribution and the corresponding loss vulnerability curve of  Typhoon-rainstorm are estimated to calculate the dynamic  hazards risk.  We use the black rectangle to denote different calculation modules and use the blue  one to represent the results obtained by the VFS-IEM-IDM model

2. **There are some unclear or awkward sentences throughout the manuscript. It is suggested that the authors conduct a thorough check to improve the readability of the manuscript.**

Thanks to the reviewer for the careful grammar checking of this manuscript. We have improved the grammar writing and the detailed can be seen in our Manuscript.

**Abstract.** Global warming has led to an  increased occurrence of compound hazards and an accurate risk assessment of such hazards is of great importance to urban emergency management. Due to the interrelations between multiple hazards, the risk assessment of a compound hazard  faces several challenges: (1) the evaluation of hazard level needs to  consider the correlations between compound hazards drivers, (2) usually only a small number of data samples are available for estimating the joint probability distribution of the compound hazard drivers and the loss caused by the hazards, (3)  the risk assessment process often ignores the temporal dynamics of compound hazard occurrences. To deal with these challenges,

**4 Case Study**

In this section, we evaluate VFS-IEM-IDM with a case study of typhoon-rainstorm compound hazards that occurred in Shenzen, China. Shenzhen is located  on the east bank of the Zhujiang River and is surrounded by Daya Bay and Dapeng Bay, where the climate is subtropical and maritime. Typhoon-rainstorms are the most frequently  occurring hazards in Shenzhen. According to the collected data, as shown in Table A1,  the direct economic losses of the Typhoon and Rainstorm hazards  from 1980 to 2016 in Shenzhen, on average, exceeded 360 million RMB per year. Also, Zhou  investigated the Typhoon and Rainstorm hazards  that caused the number of death 3.4 annually, and about 149,000 people were affected  (Zhou et al., 2017). Accurate assessments of the typhoon-rainstorm risk are crucial to determine whether or not the early warning systems are working and implemented effectively.

3. **Refer to the requirements of the journal, check and modify the formatting of the references.**

Thanks for your suggestions. We have checked and modified the formatting of the references and the detailed can be seen in our Manuscript.

**Copernicus style files**

- EndNote® Output Style File and example library
- CSL style (for Zotero, Mendeley, Papers, etc.) and example library
- Bibtex Bibliographic Style File

The risk of a hazard is defined as the potential consequences brought by the  disaster and can be quantified by the probability of losses  (He et al., 2020). Risk assessment is a technique that uses the relevant hazard data to estimate the likelihood that natural hazards may occur and further assess their economic losses  (Huang et al., 2018). Traditional methods of risk assessment mainly utilize geographic information systems to get risk maps  (Gigovic et al., 2017) or rely on information diffusion methods to deal with the problem of data sparsity  (Gong et al., 2020). These risk assessment methods  (Julia et al., 2021; Zhou et al., 2020) are mainly applied to individual hazards, while the risk assessment of compound

Alan A.: Generalized Odds Ratios for Ordinal Data. Biometrics, 36(1), 59-67, https://doi.org/10.2307/2530495, 1980.

Beaula T., Partheeban, J.: Application of Variable Fuzzy Sets in the Analysis of Synthetic Disaster Degree for Flood Management. International Journal of Fuzzy Logic Systems, 5, 153-162, https://doi.org/10.5121/ijfls.2013.3305, 2013.

Huang, C.: Principle of Information Diffusion. Fuzzy Sets and Systems, 91(1), 69-90, https://doi.org/10.1016/S0165-0114(96)00257-6, 1997.